# Reasons for Participating in the EDS-HEART Program: Holistic and Performative Within a Supportive Community

**DOI:** 10.3390/ijerph23010055

**Published:** 2025-12-31

**Authors:** Maria Kosma, Nick Erickson, Ashley L. Hinerman, Ira A. Anderson

**Affiliations:** 1School of Kinesiology, Louisiana State University, Baton Rouge, LA 70803, USA; 2School of Theater, Louisiana State University, Baton Rouge, LA 70803, USA; nickwe@lsu.edu (N.E.); iander4@lsu.edu (I.A.A.III); 3Woman’s Center for Wellness, Baton Rouge, LA 70820, USA; ahiner1@lsu.edu

**Keywords:** hypermobile Ehlers–Danlos syndrome, embodied movement program, body schema, enjoyable exercise participation, people with disabilities, phenomenology, Merleau-Ponty, love of movement, psychosocial health, public health policy

## Abstract

**Background**: Hypermobile Ehlers–Danlos Syndrome (hEDS) results in multiple, complex health-related risks and associated fear of movement (kinesiophobia). Therefore, the purpose of this research study was to examine how a holistic, embodied, and performative movement program (EDS-HEART) can affect body schema, physical and mental health, and lifestyle, which contribute to the joy of movement and physical activity participation among adult women with hEDS. **Methods**: This was a hermeneutic, phenomenological, quasi-experimental, and community-based research study among six women with hEDS, who participated in the EDS-HEART movement program at a local physical therapy clinic. The seven-week program incorporated stretching and strength training activities as well as performative-thematic movement sequences. **Results**: Based on the qualitative analysis, three themes emerged regarding reasons for the joy of the EDS-HEART program: (a) *improved body schema: body awareness, confidence, posture, and proprioception*; (b) *highly motivating program: holistic, embodied, performative, pleasant, and safe*; and (c) *psychosocial reasons: supportive setting, sense of pleasure and euphoria, and freed from social comparisons and the fear of movement*. **Conclusions:** Based on the study results, public health experts should develop and implement easily accessible and holistic movement programs among people with hEDS and similar conditions to improve physical health, psychosocial health, and the joy of movement.

## 1. Introduction

This research study is based on the philosophical underpinnings of Merleau-Ponty’s magnum opus, Phenomenology of Perception [1]. Merleau-Ponty criticized the Cartesian dichotomy of body and mind and showcased the interaction of the two via the concept of embodiment (body–mind unison). Instead of viewing the body as a physical object (like a table or a chair—being in-itself) or a statistic that can be measured, Merleau-Ponty elevated its essence to a subject, The Lived Body—with consciousness (being for itself)—that acts, walks, runs, climbs, dances, and expresses [1,2,3,4,5,6,7]. Although the body has biological, in-itself elements (e.g., cells, tissues, organs, and systems), it is not a physical object. It is an organism with consciousness (for-itself), in that there is an interplay between its in-itself biological structures and consciousness (for-itself) (see, e.g., how the habitual body connects via movement the “physical” [in-itself] with the “psychical” [for-itself] to explain the phantom limb syndrome) [1,4,8].

The body throws itself into meaningful motor significations, through which body schema or motor habit (the habitual body) is formed. Body schema reflects body awareness, posture, confidence, control, strength, mannerisms, and overall physicality. It represents corporeal expression, leading to skill acquisition and the understanding of the world [2,4,5,6,9,10]. Body schema is not a reflex or mental representation. It is a sensorimotor and pre-reflective unity whereby body and spatial awareness, posture, expression, and function are based on the integration of multi-sensorial input (e.g., visual, tactile, vestibular, and kinesthetic), especially during motricity [1,2,3,11]. These conscious and subconscious sense-giving experiences reflect *tacit* knowledge that connects the body with the world [1,2,5,9,12]. When movement skills are mastered via constant practice, praktognosia or practical knowledge is obtained, and the exerciser can perform in a state of flow whereby the execution of movement sequences (e.g., in aerial dancing) occurs without much cogitation—though with consciousness—via reliance on body schema [1,2,7,13,14,15,16]. Body schema is not static; it can constantly be reworked and renewed [1,3,9]. Holistic, performative, and embodied movement programs can lead to improved body schema, fitness, mental health, diet, the love of movement, and long-lasting physical activity participation [2,3,4,5,6,7,9,10,13,14,15].

Unfortunately, body schema can be highly imbalanced among clinical populations like people with hypermobile Ehlers–Danlos Syndrome (hEDS), leading to decreased health, impaired daily functioning, poor lifestyle choices, and diminished quality of life. About 1 in 500 to 1 in 5000 individuals are diagnosed with hEDS, which is a much more frequent condition than once thought due to increased awareness and improved diagnostic procedures [17,18]. About 70% of the diagnosed hEDS cases occur among women [19]. Based on the Ehlers–Danlos Society and recent studies, hEDS is a genetic, hereditary condition that affects the body’s connective tissue and can cause systemic bodily damage, including joint hypermobility, skin hyperextensibility, musculoskeletal pain, dislocations, scoliosis, vascular problems (e.g., stroke), cardiovascular anomalies, gastrointestinal, auto-immune, and respiratory conditions as well as autonomic dysfunction, among other health risks [20,21,22,23,24]. Chronic pain and fatigue, orthostatic intolerance (e.g., POTS: postural orthostatic tachycardia syndrome), poor sleep quality, depression, anxiety, and headaches are common symptoms of hEDS. Some of those symptoms (e.g., chronic musculoskeletal pain, high risk of dislocations and injury, and POTS) often lead to kinesiophobia (activity avoidance), resulting in deconditioning and disability exacerbation [22,25,26]. These negative health-related challenges among people with hEDS call for the identification, development, and implementation of effective exercise programs for the posited, understudied population.

Although in most exercise programs for patients with hEDS the emphasis is on strength and balance training [27,28,29], the combination of proprioception deficits and muscle weakness has called for programs within performing arts that also improve body awareness (an aspect of body schema) and decrease chronic pain [21]. It is also possible that such performative movement programs can take the mind away from the painful and debilitating hEDS symptoms, thus increasing their effectiveness, exercise adherence, and diminishing kinesiophobia. Given the vast and complex health-related risks of hEDS and the need for effective, multidisciplinary, and comprehensive treatment programs [20,30], the purpose of this study was to examine how a holistic, embodied, and performative movement program can affect body schema, physical and mental health, and lifestyle, leading to the joy of movement and physical activity participation among people with hEDS. In this way, public health experts can assist in developing effective movement programs and policies for the studied population.

## 2. Materials and Methods

### 2.1. Design and Procedures

This was a hermeneutic, phenomenological, quasi-experimental, and community-based research study among six women with hEDS (aged 26–78 years), who participated in a holistic, embodied, and performative movement program at a local physical therapy clinic. The participants’ first names were replaced with pseudonyms for result reporting. Using qualitative data in experimental designs is recommended for increased ecological validity and a more comprehensive understanding of the examined phenomenon [31]. The program incorporated warm-up activities (e.g., meditation, regulatory breathing, marching in place, walking, and light, mindful stretching) as well as strength training exercises and performative-thematic movement sequences whereby the study participants freely expressed themselves to represent the session’s theme. Some theme examples included: “camping next to a beautiful lake,” “vacationing at a favorite beach place,” “experiencing the Sahara Desert,” and “moving through space.” Given the interdisciplinary nature of the program within kinesiology and physical theater, it was named EDS-HEART (EDS-Health, Exercise, Art). The movement sequences were slow, highly controlled, and embodied. Each program session started with meditation and breathing techniques, including continuous guidance for breathing regulation throughout the entire program. The emphasis was on the connection between mind and body and the practice of sensing the movements and increasing body awareness and movement appreciation as recommended for people with hEDS [32,33,34]. Participants were able to use the wall-mounted bars for support while one of them, Emilia, performed the program mainly on the floor because of her POTS symptoms (e.g., dizziness and instability). The exercise area was covered with mats for the participants’ safety. The program instructors were the first author of the paper and two undergraduate students in the areas of kinesiology and physical theater. The sessions took place once per week and lasted between 1 h and 1:30 h. The setting was a local physical therapy clinic close to the participants’ residence. The study participants were familiar with the clinic because it was the main site for their weekly physical therapy exercises (clinic patients), where they were recruited for the study. They were living independently. The owner of the clinic, together with a physical therapist—and former student of the first study author—assisted with participant recruitment and provided access to the setting. Study participation was voluntary, and the study’s inclusion and exclusion criteria, which were approved by the researchers’ Institutional Review Board (IRB), were as follows. Participants had to be 18 years old or older with hEDS who used the physical therapy clinic (recruitment site) in the spring of 2025. Exclusion criteria were that anybody under 18 years old who had not been participating in the physical therapy clinic in the spring of 2025. Before the beginning of the EDS-HEART program, the study authors met with the participants at a scheduled “meet-and-greet” event to discuss their needs and goals. In March 2025, the two students in the study observed physical therapy sessions of the participants to familiarize themselves with their movement abilities and needs and thus assist with the development and implementation of the EDS-HEART program. The program launched on 5 April 2025, and ended on 24 May 2025. Janessa participated in all seven program sessions, while Emilia and Lauryn missed only the last session due to traveling. Lara missed two sessions because of family responsibilities. Amy and Tricia, respectively, attended three and two sessions due to medical reasons and family emergencies.

The program instructors were supportive and encouraging. They demonstrated the different movement sequences, provided verbal cues, and encouraged freedom in movement and expression. The interdisciplinary EDS-HEART program incorporated elements from kinesiology and physical theater, and it was inspired by previous embodied and performative movement programs among college students [2,3] and a community-based aerial sling class for healthy women [5,6]. It was also based on exercise recommendations for people with hEDS [32,33,34].

Data collection took place qualitatively via individual, semi-structured interviews on Zoom. Two rounds of individual interviews were performed. The first round occurred before the beginning of the program in March 2025, while the second round happened towards the end of the program in May 2025 (seven weeks later). In the first round of interviews, participants were asked about their physical activity behavior, physical therapy exercises, EDS symptomatology, physicality, and program expectations. In this way, the researchers gained an understanding of their initial exercise levels and functioning before the development and implementation of the intervention. During the second round of interviews, questions included program effects on body schema (e.g., body awareness, confidence, and mobility), physical and mental health, EDS symptoms, daily functions, and interest in continuing with the program in the future. The interview guide is shown below in Table 1. Demographic information about the participants’ age, gender, and ethnicity was also collected.

The study’s research protocol was approved by the researchers’ IRB for the protection of human subjects in research (IRB # 0081). Before the Zoom interviews, the first author electronically shared the study’s consent form and interview guide, so the participants were able to review the study’s protocol and interview questions at their own pace. Any questions regarding the research project were answered either via email communication, during the “meet-and-greet” event, or via Zoom. Prior to data collection, the participants signed the study’s consent form, and any needed clarifications were discussed with the interviewer. Study participation was voluntary; thus, study withdrawal could happen at any point without penalties or negative consequences.

The first study author conducted informal, in-depth, individual interviews in a democratic and dialogical way. The interviewer drafted the initial interview guide before discussing it with the study co-authors for any suggested changes. The final questions were then pilot tested with two interviewees, whose results were included in the study.

### 2.2. Data Analysis

The audiotapes were transcribed verbatim by a professional transcriptionist, and the transcripts were entered in the latest version of NVivo together with the interviewer’s post hoc reflections and debriefing notes. Hermeneutic phenomenology was used to analyze the study’s data [35,36,37,38,39]. Specifically, the interviewer thoroughly read the data before developing codes for themes and categories. Data coding occurred in a recursive process by relying on individual stories, the whole data set, the study’s purpose, previous experiences, and literature. Some of the results were further discussed with the study participants to facilitate the development of the study themes. The coded data were also discussed with all study authors before reaching consensus on the final coding. In this way, reflexivity and trustworthiness in this qualitative research were achieved [40,41]. As expected, in this qualitative, phenomenological, health-related, and clinical (e.g., quasi-experimental) study, purposive and smaller sampling was used to better match the study’s goal and thoroughly examine participant experiences. In this way, the study’s rigor and result accuracy also increased [42,43,44]. Representative extracts were then selected based on the coded data, the entire data set, the study purpose, and the literature.

Drawing on hermeneutic phenomenology, the in-depth examination of the studied phenomenon is imperative. Therefore, in this research, the interviewer and all study co-authors were heavily involved in the subject matter by developing and implementing the EDS-HEART program, reflecting upon current and former experiences with similar programs and research studies, continuously interacting with the participants and among each other, making program observations, and taking notes. The principal investigator used recursive analytical procedures to best capture and examine the study phenomenon. Hermeneutic phenomenology is interpretive in nature; thus, result interpretation can be flexible based on the different meanings ascribed to the shared stories [39].

## 3. Results

### 3.1. Participant Characteristics

#### 3.1.1. Demographics and Exercise Levels

Six women participated in the study: *M* age = 43 years old. Five participants were adults (age range 26–49 years old), while Tricia (the 6th participant) was an older adult (78 years old). There were four White, European American, and two women of mixed-race (White and Hispanic). During the first round of the interviews, three women were physically inactive (Emilia, Amy, and Tricia) while three others were regularly active (Janessa, Lara, and Lauryn). Based on the second round of interviews, Lara’s exercise levels increased due to the performative and holistic elements of the EDS-HEART program, as explained below in Theme 2. Excluding physical therapy, Janessa was physically active three times per week by participating in a Pilates class once per week and swimming sessions (at the physical therapy clinic) twice per week. Lauryn was very active; she exercised six days/week by participating in different classes, including Zumba, Pilates, Barre, and swimming.

#### 3.1.2. EDS Symptoms and Comorbidities

All participants were diagnosed with hEDS and/or hypermobility spectrum disorder. Accordingly, nearly all participants experienced chronic musculoskeletal pain in hands, wrists, shoulders, knees, ankles, and lower back (e.g., pelvic floor); poor proprioception; kinesiophobia (fear of movement); digestive issues; and depression. Tricia was also diagnosed with Atrial Fibrillation. Other hEDS comorbidities included Raynaud’s Disease and dysautonomia (for Lara) and eosinophilic esophagitis for Janessa. Most of the participants followed a special diet, including avoidance of processed and fried food, sugar, and dairy. Two of the participants were pescatarian.

### 3.2. Emerging Themes

Three themes emerged regarding the multiple reasons for loving and participating in the EDS-HEART program, including participants’ willingness to continue with the program should it be offered in the future. These themes included: (a) *Improved body schema: Body awareness, confidence, posture, and proprioception*: the embodied, thematic and performative elements of the program coupled with the breathing techniques led to increased motivation and improved body schema; (b) *Highly motivating program: Holistic, embodied, performative, pleasant, and safe*: the embodied, performative, pleasant, and gentle movement sequencies were highly motivating and helpful to return to regular exercise classes; and (c) *Psychosocial reasons: Supportive setting; sense of pleasure and euphoria; freed from social comparisons and the fear of movement*: the participants learned from each other within a supportive community freed from social comparisons and kinesiophobia. A summary of the emerging study themes with selected participant sample quotes can be viewed below in Table 2.

#### 3.2.1. Theme 1: Improved Body Schema: Body Awareness, Confidence, Posture, and Proprioception

The participants indicated that they improved body awareness, posture, confidence, and proprioception because of the holistic, embodied, and performative movement sequences. Also, these gentle and highly controlled exercises assisted with swiftly and smoothly returning to regular exercise after stopping due to sickness. The performative, embodied, controlled movements of the program, together with the breathing techniques, improved body awareness, balance, proprioception, sensations of fluidity, and excitement about being physically active.

#### 3.2.2. Holistic, Thematic, Embodied, and Performative to Improve Body Schema and Return to Exercise

Although there was a consensus that meeting twice per week would have been more beneficial than once per week (schedule conflicts did not allow for this to happen), Lara described the EDS-HEART movement program as “phenomenal,” where body and mind connect, leading to improved body awareness and confidence.

“I liked the combination of the two (e.g., strength training and performativity)… we kind of actually feel like we did three… we stretched and talked about how everybody felt and the direction we were going. And then we went into like strengthening and… having conversations about different muscle stretchers. And then we got moving and laughing. Let me tell you… towards the end… game where… You had to be still—we had to come together to be an object… I thought that was really interesting because, again, it’s making yourself aware… It’s a transition experiment. ‘All right, let’s move, but let’s figure out how to put some attention on some of these muscles to get them, like, actually addressed.’ I genuinely feel like EDS people; our brains do not address anything unless we mentally address it. Sometimes I feel like I have to genuinely tell my knees: ‘Okay, wait. No. You’re supposed to do this.’ Or like when I go to pick something up, I’m like, ‘Okay, we’re not using all of this. We’re using these.’ So, getting body awareness through the strengthening and then transitioning into, ‘let’s get moving’ and like, ‘doesn’t this feel better?’ Like lighthearted and then you know, walk out of there feeling really good and confident”.(Lara)

Amy is now “moving more” and, due to the EDS-HEART program, she is “paying attention to her body (e.g., upper body)” when she is “walking around the house.” Tricia, Amy’s mother, also enjoyed all aspects of the program, from the performative themes (e.g., “pretending they were in a jungle”) to the “different and holistic ways of moving, stretching, adding music, and correcting posture.” Lauryn, who is very active, noticed improvements in her body posture and appreciates the use of mirrors to assist with the correction of body positioning. She explained that people with EDS tend to have an imbalanced proprioception, and they need cues to differentiate between a healthy posture and a comfortable but harmful one.

“…having just one more opportunity to work on it is always helpful. Like, I really appreciate that the city has mirrors. I can check my posture. I must be aware of it cause when we go for a walk around, I see myself, and so I can make quick modifications to things. And so, any input… is always appreciated… because your joint, I think, goes to like too far a range of motion, and that feels normal to you. You’re more likely to hang out in a harmful posture. So, for example, we talked about things like locking our knees. If I listen and bend my knees to what looks normal in a mirror—they look straight in the mirror—but to me, it feels like I am bending my knees like a 30-degree angle or something… because normally this feels comfortable”.(Lauryn)

Lauryn is very active; thus, she did not notice major changes in her range of motion or mobility due to the EDS-HEART program—like the others, she mentioned that meeting twice per week could be more beneficial for strength and mobility. Nevertheless, the gentle, highly controlled, and embodied movement sequences of the EDS-HEART program were key to easily returning to her regular exercise classes after having to stop exercising for a week.

“So, for the most part, I would say no (change in range of motion or mobility) because I’m generally pretty active anyway. However, I don’t always do the type of stretching that we do in class, so it’s always helpful for me to do a little bit of stretching. And this would create an opportunity for me to pursue that. When you first sent out the questionnaire, and I saw this question, I was going to say for me, absolutely not… it just stayed the same… But last week I was actually really sick, and I didn’t go to any movement classes at all. And yesterday was the first time I moved in like a week (due to EDS-HEART). And it definitely helped to kind of just lubricate joints a little bit more, take things very gently. I was a little bit nervous about getting back into doing the extensive movement that I do in like barre classes, or the amount of cardio for a full hour of Zumba. And so, this was very—yesterday was very reassuring, so I was able to go to a barre class today because taking a full week off, being in pain, makes it really challenging to like—every time I take a week off, it’s like it takes three weeks to get back to where you are. So, gentle movement was very helpful for me in this case. And so, I think I conclude from that that for people who are much less active than I have been in recent months, that you’d probably see a greater effect size in terms of this class and how it makes me feel”.(Lauryn)

Although Janessa finds the open-chain physical therapy exercises (e.g., using the “cable machine” or doing “squats and calf raises”) important to strength training, she mentioned that the closed-chain exercises of the EDS-HEART program are more effective in “building stability and proprioception, which is also supported in the literature.”

#### 3.2.3. Breathing Techniques for Improving Body Schema

Lara learned how to breathe during exercise due to the holistic and performative nature of the EDS-HEART program, in which breathing techniques were incorporated. She did not realize “how much she was holding her breath during exercises” until she started participating in the study’s program. Instead of focusing only on the use of physical therapy equipment for her injured knees, she learned how to also concentrate on her breathing, which brought up a feeling of “fluidity” and the need to “listen to her body.”

“And I didn’t realize that for the majority of my movements I’ve been holding my breath… This (EDS-HEART program) is a shift. I have been so grateful! This has been one of the greatest blessings! This has been really good! It’s motivating; it’s shifted my way of thinking about physical movement, exercise, how it’s supposed to feel too… feeling good… like coming in feeling really tight, rigid, and just awkward, and then leaving just feeling so fluid. And I think breath has to do with that; learning that it doesn’t have to be a certain way; there’s not a wrong way to do it, as long as you’re listening to your body. And the performative aspect helps you get in touch with your body in a way that exercise doesn’t”.(Lara)

Janessa also praised the importance of breathing exercises to stabilizing and sensing her “core and pelvic floor,” with which she has issues. She particularly enjoyed the holistic, controlled, and thematic movements: they were enjoyable and motivating, leading to improved balance and proprioception—in a fun way—without worrying about the exact position of her body.

“…mentally, I do enjoy the themes. I think they’re fun. I will say two of the highlights for me were the last session, climbing up and down the rocks. You know, that was a movement that used our whole body. It was balance-oriented since we were lifting one leg at a time, but not for so long that any of us toppled over, and the cue of like going up the rocks and then coming back down, you know, makes you key into how your body would move differently doing that same move in different directions. That one was a winner for me, and that’s an exercise I would hate doing if we weren’t doing it with some kind of purpose, but it’s a valuable one. And then I’m trying to remember the exact move, but one of the ones from our journey to the sacred tree, where the student was really focused on slow, careful, exaggerated movements. I don’t remember if we were stalking jaguars or getting through a tunnel at that point. But there was a move where it was like a very slow, concentrated—you know, we’ve got to move very deliberately. And those are really valuable for me because it gives me the time to stop and think... ‘I lift this foot and how far does it have to go?’ while also still being kind of fun and diverting. So, there are some glimmers where it’s like, ‘oh, this is, you know, this lets me do something really valuable and be distracted from how worried it can be to think, you know, ‘My foot has to go exactly here’”.(Janessa)

### 3.3. Theme 2: Highly Motivating Program: Holistic, Embodied, Performative, Pleasant, and Safe

All participants expressed enjoyment of participation in the EDS-HEART program. It was highly motivating due to its holistic, thematic, performative, and embodied nature. The movement sequences were controlled, pleasant, and safe, reinforcing free expression.

#### 3.3.1. Holistic, Performative, Embodied, Enjoyable

“…it’s just a different feeling though… It’s a different motivation… There is something like lightheartedness. It’s not like ‘I am trying to get to where this weight that I’m pulling on this machine is not feeling as difficult when I started it at the beginning of the week.’ Like it’s more of just whole-body, I guess, and that’s the performative aspect, it’s like everything moving versus just a single muscle group kind of thing… the fun, performative aspect kind of helps motivate you. Like, this is not just some ‘all right, I’m gonna just move my arm.’ There’s definitely a different feeling than physical therapy for sure… I’d go in there (physical therapy), I’d do stairs, I’d do tires, I’d kind of be… ‘I’ve got a lot to do, this, that and the other’… It’s not just like you know that you’re physically trying. Like in physical therapy, there are specific goals. But in this, I feel it’s less goal-oriented and more like ‘How do I feel today and how do I get my body moving’ and like the least-boring… It’s not like I’m having to make myself do it… There’s an element of it that like, ‘yeah, I know that it’s gonna be tough to get moving again’… It’s just a different feeling though, cause it’s a different motivation… having that time and space dedicated to it and then kind of just like going into it, I always felt better leaving”.(Lara)

“It’s very different from physical therapy, cause physical therapy’s still targeted for a particular body part. I think for overall health, there needs to be a more holistic approach, like we were doing here, in terms of approaching movement. But because of that, they’re not interchangeable… of course, that was never the intention, but I would need a very structured approach to like rehabbing my shoulder, for example”.(Lauryn)

Similarly, Lara specified that contrary to physical therapy, whereby the emphasis was on rehabilitating her injured knees after her car wreck (“injury-focused” approach), in the EDS-HEART program, the focus was on the “whole body without feeling overwhelmed.” She started learning “how to move again” and has a good sense of her body and movements—“reconnect with her body.” Most importantly, she increased her exercise levels by being active at home. She imagines, creates, and incorporates performative and thematic movement sequences with music (like the nature of the EDS-HEART program). Movement to Lara is not “dreadful” anymore (e.g., “I need to do 8 leg raises, 8 clam shells, and dread it”). Her perception of movement dramatically shifted by finally feeling comfortable and enjoying being physically active in an imaginative and performative way (e.g., during a rainy day).

#### 3.3.2. Pleasant and Positive Without Negative Impact; Gentle Way to Prepare for Other Exercise Classes

Lauryn and Janessa also mentioned that the program was pleasant, safe, and positive without causing any injuries or negative impact.

“It’s a very safe space. It’s very low impact, and of course, I felt very comfortable that if I needed to not do something, I could. There is no problem with looking for modifications for certain movements. And we never did any particular movement for such a long time that it could increase the risk of injury… with EDS, where it’s repetitive motion that causes wear and tear on tendons. So that it felt very risk-free”.(Lauryn)

“I will say, as we get through class without pain or injury most of the time… and that’s not always a guarantee with an exercise class for me. Which makes it difficult to find enough exercise to keep me happy with getting enough exercise, so there’s not a reduction in symptoms, but there’s also—with the exception of sometimes overstretching stuff—not an aggravation in symptoms, which isn’t always guaranteed when I’m doing physical activity”.(Janessa)

### 3.4. Theme 3: Psychosocial Reasons: Supportive Setting; Sense of Pleasure and Euphoria; Freed from Social Comparisons and the Fear of Movement

All study participants were excited about the EDS-HEART program. It was motivating to be around other people with EDS and learn from each other. Being able to exercise together and share similar struggles and different ways of moving gave them strength and motivation to continue with the program. The participants felt comfortable without thinking about social comparisons or fearing movement, which they typically experience in regular exercise classes due to their competitive and goal-oriented nature. The EDS-HEART program was different from physical therapy, making it highly motivating vs. “boring or dreadful.” The movement program was also “mentally very pleasurable” and gave a sense of euphoria.

#### 3.4.1. Supportive Setting, Learning from Each Other; Positive Feelings

Emilia said that the program was interesting and engaging within a small group of people, contrary to the larger groups in physical therapy, so they could ask questions, “get more attention,” and learn from each other. She finally learned how to “tuck her rib cage in” and “move safely.” Janessa also benefited from “the tip” regarding “pulling the rib cage in.” Similarly, Lauryn explained that by interacting with the other participants in the class, she found out why, at times, “mobility is blocked due to the clicking motion in some of her joints (e.g., hip motion) and how to deal with it.”

“But I feel like since there is like a group aspect and it’s not just like you by yourself, like doing the movements and like trying to make your body work… It’s not as, I guess, measurable a process. We’re moving, we’re laughing, we’re talking… You can ask questions about what you’re supposed to be working on and how something feels. And you have somebody who has your body structure, who has the same body, that can be like, ‘I feel that same exact thing.’ And then you’re like, ‘Okay, well, how does it help you?’ Or ‘how do you get that muscle to start working?’… It’s like dialogue, I guess, that helps you keep moving despite how you’re feeling” .(Lara)

“…being able to have not just like the security of like somebody who’s trained like a physical therapist to help you be able to do these movements and feel confident and safe doing so, or even to like get more of a stretch or strength… we have that, but then… You also have somebody who has your body structure, and so having somebody to be like, ‘yeah, I know… my ankles do that thing too’… watching them like try to move their ankles and like get their structure also kind of helps you visualize… cause not everybody has as much flexibility”.(Lara)

“… the mental health aspect cannot be understated… oftentimes, they have the ribbons that represent different awareness. Well, the ribbon that’s for EDS is actually zebra printed, and the reason why the zebra is chosen… is because it is very hard to find people who have an actual diagnosis, and then you know, who can find that sense of community… we are just kind of like a rare breed. But when we like come together, we realize that there is an element of being stronger because we’re not alone. It’s almost like you get locked in your own sphere of movements when you go through different phases of pain and limitation. But being able to see people in different phases of their different journeys with their different joints and different things like that, it helps give you strength to like push through what you’re experiencing”.(Lara)

“…it can sometimes feel like a lonely road… our spatial awareness and the way we can move and do move is not always fluent. So being around other people who move like us and then adding that level of performativity on top of it, there’s a certain level of almost a comfort because we all move the same… if we were in like a dance class, and there’s just the level of, ‘I know that I’m really awkward, and when I bend my arm it goes a little bit farther than everybody else’s…’ it’s less comparative, it’s a level of almost like comfort, and I feel like I haven’t had in a normal setting that kind of motivates me in my everyday life to be like, ‘hey, I may feel like this. I may be the one with health issues, but I’m not the only one experiencing this. And if I really need something… I know that come Saturday, like I’m gonna be with my people, and I can talk to them about it…’ It’s a different element that I didn’t expect to have, being able to feel that comfortable moving, like performative wise”.(Lara)

Similarly, Lauryn and Janessa enjoyed exercising together with other EDS patients. They could share similar struggles and exercise techniques that may work for them. They can learn from and help each other.

“It is emotionally and mentally very pleasurable because it’s a mixed group of EDS patients. And that’s an experience that we never get. Like, we don’t get to hang out with each other, especially in fitness classes. So, there is a real benefit on that level, just being able to talk to each other and work out with each other. To not be the only person in class going, ‘I can’t do it that way. Can we figure out a different way to do this?’ So, there’s that, and there’s the general euphoria of having exercised. The social aspect has been lovely… It’s huge to be able to say, ‘oh I got a new cervical traction machine. Does anyone else have it? How does it work for you? How do you like this doctor? Does this technique work? Which electrolytes don’t taste awful?’ That’s massive”.(Janessa)

#### 3.4.2. Freed from Social Comparison and Fear of Movement

Lara also emphasized that the performative aspect of the program within the supportive setting among only EDS people takes the mind away from social comparisons and how certain body parts are supposed to move. It also removes the fear of overstretching a muscle or doing something wrong compared with others in a regular class who do not have EDS. Instead, the emphasis is on feeling the movement and what it represents.

“…we’re comparing ourselves to somebody who doesn’t have the same abilities. That’s the other thing. We are in a workout class, for example… you’re gonna have a mixed group of people and abilities… but there’s like a real difference when it comes down to their limitations and ranges of motions, and then like our ranges of motions. It’s like a different kind of focus. Because it’s like other people don’t have to often think about, ‘hey. Just because I can throw my arm this way doesn’t mean I should throw my arm this way.’ And people are like, ‘lean into the stretch.’ And I know that they mentioned overstretching and things like that… this has helped me figure out how to not run away from how my body feels, but to listen to my body… Cause I feel like when you put your mind on just how your body moves, and not moving your body that far or not squeezing this—or trying to squeeze this muscle or engage this muscle while also breathing, while also keeping your back straight, while also doing this, it’s so much… I’m gonna just take a huge deep breath, and I’m gonna be a giant tree… it separates you from fear and comparison. And then the feeling like you’re not doing it right. It takes the right kind of movement out of it. It makes it about how it feels and then what you want it to feel like and what you’re trying to represent… Instead of being like…. ‘Reach your hand up high and then you’re gonna lean to the left, and you’re gonna lean to the right’… and you’re thinking about that, you’re like in your head going through so many movements… ‘I don’t want to stretch my arm out too far here, and I don’t want to look like this’… Or ‘this feels weird. They’re moving their body like this. If this obviously doesn’t hurt them, but it’s hurting me.’ Like, there’s so much of a mind going on in your mind and then trying to connect with your body, whereas like the performative aspect, it’s like ‘all right. I’m not thinking about my body. I’m gonna think about being a giant tree. I’m gonna stretch my limbs up high, and then I’m gonna stretch them down low, and then that’s it… And it allows you freedom, like not comparing”.(Lara)

Lauryn mentioned that at times the performative aspect of the EDS-HEART program is a “reprieve, taking the mind away from pain,” like when she had a “toe issue” and she “was worried about having to schedule more physical therapy sessions or seeing another doctor.”

“I think mentally it gives me a little bit of a reprieve. So, for example, I can think of one session where I came in experiencing a particular problem, and it took a little bit for me to be able to get my mind off that problem. So, it gave my body kind of like a way to just like not hyper-fixate on pain for an hour. And then when I finished class, I came back and could approach the pain again, but from hopefully a place of a little bit less anxiety about it. I think I wasn’t in like horrendous pain. It’s hard to get right off it when it’s all-consuming. But for moderate to mild pain, it was very helpful. The way like watching a TV show or something, but while still doing movement, that’s a really helpful way to kind of take a step back from anxiety, is that distraction, getting involved in the story, wondering like what step or movement are we gonna do next? Almost kind of like meditation, but more to me like a dance class”.(Lauryn)

## 4. Discussion

This is the first study to examine how a holistic, embodied, performative, and community-based movement program (EDS-HEART) can affect body schema, physical and mental health, and lifestyle, factors leading to the joy of movement and physical activity participation among patients with hEDS. Each theme will be discussed separately in relation to the study’s philosophical framework, the literature, and participant excerpts. Study strengths, limitations, implications, future directions, and conclusive comments will also be reported.

### 4.1. Theme 1: Improved Body Schema: Body Awareness, Confidence, Posture, and Proprioception

Based on the first study theme, not only did the EDS-HEART program lead to improved body schema, balance, and proprioception, but it also created excitement about continuing to be physically active. It also assisted with easily and quickly returning to regular exercise after stopping due to sickness. The positive effects of performative and embodied movement programs in performing arts on body schema among mainly healthy college students and women were recently supported [5,9]. Although most studies have examined how body awareness and proprioceptive training improve sensorimotor function and mental health among different clinical populations [45,46], uniquely in this study, the breathing techniques implemented throughout the EDS-HEART program also improved body schema.

These findings are in alignment with Merleau-Ponty’s philosophy in that an imbalanced body schema—which is common among clinical populations like people with hEDS—can be improved via controlled, thematic, performative, and embodied movements that also incorporate breathing exercises. Constant practice of embodied exercises can lead to improved sensorimotor function by better connecting movement to the multiple sensory systems responsible for body posture, awareness, and functioning. This process is not linear (stimulus-response) or mechanistic because the body is a subject (with consciousness) and via body schema it can link its in-itself structures and for-itself elements [1,3,4,8]. Such embodied and performative movement experiences can also be enjoyable, highly motivating, and lead to the ease of returning to regular exercise after movement ceases due to sickness or injury. In studies for people without disabilities, it has been shown that embodied and performative movement programs can increase not only body schema but also the love of long-lasting exercise participation [4,5,6,9].

### 4.2. Theme 2: Highly Motivating Program: Holistic, Embodied, Performative, Pleasant, and Safe

Nearly all participants emphasized major differences between the EDS-HEART program and physical therapy, in that the embodied, thematic, and performative program was highly motivating and enjoyable. It was not boring or drudgery like with physical therapy. It was something to look forward to and inspiring to incorporate similar thematic movement sequences at home. The program was also safe and pleasant without causing any injuries or over-stretching, which is typical among people with hEDS during exercise. Although similar movement programs have shown to lead to the love of movement among non-disabled populations [6,14,15], this is the first study to showcase that holistic, embodied, and performative movement experiences can eliminate fear of movement or kinesiophobia—and its underlying negative health consequences—which is common among people with hEDS [22,25,26].

Expanding upon Merleau-Ponty’s philosophical underpinnings on embodied movement, a potential reason for this positive effect may relate to improved body schema (e.g., having a better sense of one’s body and functions) that links not only to improved functioning but also to joy and confidence in one’s physical capabilities [6]. The performative and thematic elements of the program probably also relate to sensations of play, whereby the emphasis is on creating and sharing a story (playful elements of performing arts) and thus taking the mind away from worries such as fear of movement [47,48].

### 4.3. Theme 3: Psychosocial Reasons: Supportive Setting; Sense of Pleasure and Euphoria; Freed from Social Comparisons and the Fear of Movement

Another reason for diminishing kinesiophobia in the study was the social element of the EDS-HEART program. Contrary to regular exercise classes, study participants had the opportunity to exercise together with other hEDS patients and form a community. Social comparison decreased, and the participants felt comfortable exercising and learning from one another. The social bonding also contributed to positive feelings of “reprieve” and “euphoria.” The tendency to form strong and supportive communities via performative and embodied movement programs is shown in other studies among non-disabled populations [9,10,13].

Drawing also on Merleau-Ponty’s philosophy, social interactions and emotions are embodied, in that they are influenced by body schema [3,4,9,10,13,14]. The importance of embodied movement to emotional expression and social interactions contradicts the Cartesian body–mind dualism and reinforces the pre-reflective nature of our consciousness and socio-emotional expressions [4,5,9,10,49,50,51].

This is the first quasi-experimental phenomenological study to examine how the embodied, thematic, performative, and holistic elements of the interdisciplinary EDS-HEART program led not only to improved body schema, functioning, and psycho-social health but also to decreased kinesiophobia and the joy of movement within and outside the program’s setting. Another study strength involves the inclusion of a clinical population with both similar and diverse characteristics (e.g., musculoskeletal pain, fatigue, POTS, depression, and digestive issues) in a familiar physical therapy community setting. Although the participants agreed that meeting twice per week would have been more beneficial than once per week—especially due to the combination of exercise training and performativity—this could not be achieved during this pilot EDS-HEART program due to participant schedule conflicts and limited resources. Even though the study’s smaller sample size may be perceived as a study limitation, in qualitative, phenomenological, health-related, and clinical (e.g., quasi-experimental) studies, purposive and smaller sampling is recommended and expected to better match the study’s goal and thoroughly examine participant experiences. In this way, the study’s rigor, and result validity and reliability can also be supported [35,36,37,38,39,42,43,44]. The smaller nature of the class was praised by the participants in that it was comfortable and effective in improving and promoting positive social interactions, like learning from one another. That said, longer interventions and mixed-methods designs may be necessary to further validate the results of the current study. Given the high prevalence of hEDS among women and the fact that mostly women with hEDS were involved with the physical therapy clinic, it is not surprising that all study participants were females. Although the study participants were somewhat ethnically diverse and one of the students administering the program was an African American male, in future studies, a more diverse group of people (e.g., inclusion of African American men) should be examined.

Based on the results of this study, certain implications can be proposed for exercise promoters and public health officials. Exercise trainers should incorporate holistic, embodied, performative, and breathing elements in their movement programs to increase body schema (e.g., a good sense of the body and its functions), exercise confidence, and the joy of movement. The physical activity setting needs to be supportive in nature, preferably involving people with similar conditions (e.g., hEDS) to learn from each other, reinforce positive feelings within a safe space, and thus decrease kinesiophobia and social comparisons. Public health officials need to provide access to similar exercise programs within the community so that people with hEDS and other clinical conditions can have the option to exercise with their peers in a comfortable and affordable setting. A multidisciplinary involvement is encouraged, including experts in kinesiology, performing arts, public health, and physical therapy.

## 5. Conclusions

This was the first phenomenological and quasi-experimental study providing initial support of program effectiveness, where an embodied, performative, and socially supportive movement program (EDS-HEART) within a community setting led to decreased Kinesiophobia and the joy of movement via improved body schema and psycho-social health. Given the pilot nature of this study, future research is needed to examine repeatability. Kinesiologists, movement educators, and public health officials are encouraged to develop and implement easily accessible and holistic movement programs among people with hEDS and similar conditions. Individualistic physical therapy can be rigid; thus it is important to supplement it with embodied and performative movement programs within supportive community settings. In this way, their effectiveness can improve not only from a physical or psycho-social aspect but also from a motivational perspective, potentially leading to the long-lasting joy of exercise participation among people with disabilities and clinical populations, which is a public health priority.

## Figures and Tables

**Table 1 ijerph-23-00055-t001:** Interview Guide.

*Round 1 (Pretest) Sample Questions (Prior to EDS-HEART Program)*
1. What is your diagnosis, and for how long have you been diagnosed with this condition?
2. What are your EDS symptoms and major struggles?
3. For how long have you been following physical therapy (PT) at the clinic?
4. Beyond PT, are you physically active? In what types of exercises do you currently participate, how frequently, and at what intensity?
5. Has PT helped you with your symptoms and how?
6. Have you faced any challenges with PT? Explain.
7. Which PT exercises have been helpful and how/why?
8. Have you ever participated in performing arts like dancing or physical theater?
9. What is your perception of your physicality, like body posture, awareness, daily functioning, confidence, and expression?
10. Do you have any health problems, including physical and mental health issues?
11. What are your expectations of the EDS-HEART program?
12. Describe your diet.
** *Round 2 (post-test) questions* **
1. How has the EDS-HEART program affected your EDS symptoms?
2. How has the EDS-HEART program affected your physical health?
3. How has the EDS-HEART program affected your mental health?
4. How has the EDS-HEART program affected your physicality (body schema) in your daily activities and functions?
5. How has your body movement and function changed in your daily activities?
6. How do you view your body in the movement program and daily life?
7. How do you view your body movements (e.g., physicality) in the program? Did they change and how?
8. Can you describe an example in class to show how you get a good sense of your body, body movement, function, and expression?
9. Do you have any health problems?
10. How has the EDS-HEART program affected your emotions? Think of how your bodily exertion in class affects your feelings.
11. Has your diet changed, and how?
12. Have you met your class expectations, and how?
13. What would be some program-class changes that you would recommend?
14. Did the class help you achieve or rethink some of your life goals, and how?
15. Do you want to continue with the movement program in the future?

**Table 2 ijerph-23-00055-t002:** Emerging Themes Summary.

Emerging Study Themes—Reasons for Adherence to the EDS-HEART Program	Summary with Selected Participant Quotes
***Theme 1***: Improved body schema: Body awareness, confidence, posture, and proprioception	“…liked the combination of strength training and performativity… having conversations about different muscle stretchers… let’s figure out how to put some attention on some of these muscles to get them like actually addressed.”
	Enjoyed all aspects of the program, from the performative themes (e.g., “pretending they were in a jungle”) to the “different and holistic ways of moving, stretching, adding music, and correcting posture.”
	Gentle, highly controlled, and embodied movement sequences were key to easily returning to regular exercise classes following cessation of exercise.
	Fluidity in breathing instead of holding one’s breath.
	Pretending to “climb up and down the rocks… balance-oriented…. slow, controlled, concentrated, deliberate movements” for increased balance and proprioception.
***These 2***: Highly motivating program: Holistic, embodied, performative, pleasant, and safe	“…it’s more of just whole-body, I guess, and that’s the performative aspect, it’s like everything moving versus just a single muscle group kind of thing… the fun, performative aspect kind of helps motivate you… There’s definitely a different feeling than physical therapy… in this, I feel… like ‘How do I feel today and how do I get my body moving’ and like the least boring.”
	“It’s very different from physical therapy, cause physical therapy’s still targeted for a particular body part. I think for overall health, there needs to be a more holistic approach like we were doing here in terms of approaching movement.”
	“Whole body and performative movement without feeling overwhelmed.” Movement is not “dreadful” anymore. Change in perception of movement by finally feeling comfortable and enjoying being physically active in an imaginative and performative way.
	The program was pleasant, safe, and positive without causing any injuries or negative impact: “We get through class without pain or injury most of the time… which isn’t always guaranteed when I’m doing physical activity.” “It’s very low impact, and of course I felt very comfortable… very risk-free.”
***Theme 3***: Psychosocial reasons: Supportive setting; sense of pleasure and euphoria; freed from social comparisons and the fear of movement	The program was interesting and engaging within a small group of people—contrary to the larger groups in physical therapy—so they could ask questions, “get more attention”, and learn from each other.
	“We’re moving, we’re laughing, we’re talking… You can ask questions about what you’re supposed to be working on and how something feels. And you have somebody who has your body structure, who has the same body… ‘I feel that same exact thing.’ And then you’re like, ‘Okay, well, how does it help you?’ Or ‘how do you get that muscle to start working?’
	“…the mental health aspect cannot be understated… being able to see people in different phases of their different journeys with their different joints and different things like that, it helps give you strength to like push through what you’re experiencing.”
	The EDS-HEART program is a “reprieve, taking the mind away from pain.”
	“…it’s less comparative, it’s a level of almost like comfort, and I feel like I haven’t had in a normal setting that kind of motivates me in my everyday life to be like, ‘hey, I may feel like this. I may be the one with health issues, but I’m not the only one experiencing this… It’s a different element that I didn’t expect to have, being able to feel that comfortable moving, like performative-wise.”
	“…it separates you from the fear and comparison. And then the feeling like you’re not doing it right. It takes the right kind of movement out of it. It makes it about how it feels and then what you want it to feel like and what you’re trying to represent… the performative aspect, it’s like ‘all right. I’m not thinking about my body. I’m gonna think about being a giant tree. I’m gonna stretch my limbs up high, and then I’m gonna stretch them down low, and then that’s it… And it allows you freedom, like not comparing.”

## Data Availability

The original contributions presented in this study are included in the article. Further inquiries can be directed to the corresponding author.

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
