# Peer review of "Reasons for Participating in the EDS-HEART Program: Holistic and Performative Within a Supportive Community"

_ijerph, 2025, doi:10.3390/ijerph23010055_

Round 1

Reviewer 1 Report

Comments and Suggestions for Authors

This manuscript explores the possible use of an interdisciplinary movement program in individuals with Hypermobile Ehlers-Danlos Syndrome.

Though the originality of the intervention, the study shows some limitations that should be acknowledged and declared by the Authors.

First, though it could be considered sufficient for a qualitative pilot study, the small sample size, especially considering the inter-individual differences, should be mentioned. Moreover, the length of the intervention, especially considering the limited number of sessions per week, is too limited to explore participants' experience in the long term. Furthermore, the qualitative design of the study does not allow to draw definitive conclusions, such as those reported. I suggest to revise the conclusions section by use more cautious terms regarding the future employ of similar programs in this type of patients. Quantitative, longer research on wider samples are needed to support these findings.

I also suggest the Authors to consider changing the title as follows, or in a similar more explicative way: Reasons for participation of patients with Hypermobile Ehlers-Danlos Syndrome in a holistic, embodied and performative movement program within a supportive community.

Finally, the Authors exceeded the limited number of self-citations established by journal requirements (https://www.mdpi.com/ethics#_bookmark20). I'm aware that the Authors experienced movement programs to improve body schema in several previous studies, but they should reduce the number of self-citations by reporting only those that are fundamental in designing the background for the present study. 

I think that with these changes the manuscript could be considered for publication.

Reviewer 2 Report

Comments and Suggestions for Authors

There is no need for the keywords to repeat the information from the article title.

In the introduction section, the Merleau-Ponty concept of embodiment is explained, citing Phenomenology of Perception and eight articles by the first author of the reviewed article.

The first author appears as a character in the research presentation and autocites herself 15 times, accounting for 37% of the references. It potentially undermines the analysis's perceived objectivity, besides the journal discourages excessive autociting. 

The research intervention involved a vulnerable group of six women, who present symptoms such as chronic pain and fatigue, orthostatic intolerance, poor sleep quality, depression, anxiety, headaches, joint hypermobility, skin hyperextensibility, musculoskeletal pain, dislocations, scoliosis, vascular problems, respiratory conditions, kinefobia, and digestive issues. After "nearly" seven sessions for 4 of those women and fewer for the other 2, the use of the expression "love of moving" as a result of the intervention seems unrealistic. Adherence or joynment would be more appropriate.

It is necessary to specify how many women attended the intervention sessions and how often, out of the 42 possible (6 women x 7 sessions).

The interpretive analysis (hermeneutics) of this article's author(s), the variable sample sizes, and women's participation cast doubt on the articles' scientific rigour.

It was presented a table with preliminary and final question that are not clearly addressed. A table with the main themes, the subseqent responses and comments to sustain the research objectives would be more precise and easy to follow.

Reviewer 3 Report

Comments and Suggestions for Authors

Dear authors,

The manuscript is very interesting, as there are no existing studies on Hypermobile Ehlers-Danlos Syndrome that address the topic from this perspective — proposing physical exercise sessions that emphasize the connection between body and mind, and employing a qualitative approach. This methodology allows for results that would be difficult to obtain quantitatively, given the significant challenges faced by these individuals.

However, there is a high number of self-citations. Although it is clear that you are a leading reference in this field and in this type of exercise-based approach, you should carefully select which citations are truly essential, rather than including such a large number of references for individual ideas. For example, on line 39, nine works are cited. While all of them may be related to the idea being conveyed, some are likely more directly relevant. The same applies to the citations on lines 49, 54, and 61. Conversely, on line 78, additional references should be included to reinforce this idea and to guide readers toward other studies involving physical exercise interventions with this population.

Regarding the type of study, I understand that you define it as hermeneutic and phenomenological, but not quasi-experimental. Why are these latter characteristics included if no quantitative research was conducted?

Concerning the names mentioned on lines 105 and 121, I assume they are pseudonyms. Although this is later clarified on line 148, this information should appear earlier in the text.

Finally, on line 179, the average age is presented, and two lines later, the age range. It would be preferable to present all this information together for clarity.

Best regards.

Reviewer 4 Report

Comments and Suggestions for Authors

Dear Authors,
the idea of art therapy is a valuable insight into the possibility of treating people with Hypermobile Ehlers-Danlos Syndrome.

In my opinion, however, the article itself contains serious errors in the design, implementation and writing of the article:

  • No clear research objective
  • No information on where the research subjects were recruited from
  • Very small study group (n=6)
  • Wide age range (26–78 years)
  • Training with stretching elements in a group of patients with excessive mobility (line 94) is not recommended
  • No information about bioethical aproval - form of research experiment. There is information about the approval of the research protocol. I do not understand if this is the same thing?
  • No reliable assessment of participants before and after the intervention, which does not provide reliable results. The authors write about a reduction in kinesiophobia without using an appropriate tool (e.g. Tampa Kinesiophobia Scale)
  • No inclusion and exclusion criteria for the study were provided
  • The results are not clear and concise
  • Conclusions too general, no confirmation in the results

In my opinion, the article is not ready for publication in its current state. The methodology should be reviewed and the research repeated. Perhaps a journal in the field of humanities would be a good place to share your project. 

Best regards

Round 2

Reviewer 2 Report

Comments and Suggestions for Authors

There are at least 3 articles addressing the same side of the subject: body schema or aerial practice. 
Reducing those titles to the most representative for this article in review seems more appropriate than reducing the self-citation percentage by adding authors to the reference list.

Author Response

See attached file regarding my response.

Reviewer 4 Report

Comments and Suggestions for Authors

Dear Authors,
thank you for the corrections you made.
Please refine a few things:

1. There is still no information about where the study participants were recruited from. For example, were they hospital or clinic patients?

2. Training with stretching elements in a group of patients with excessive mobility (line 94) is not recommended.

You provided a reference to the association's website. Please cite the reviewed article.

3. Inclusion criteria not clear. Exclusion not included.

Author Response

(The authors gave the same response as above.)
